

# Solution strategy based on Gaussian mixture models and dispersion reduction for the capacitated centered clustering problem

Santiago-Omar Caballero-Morales

Postgraduate Department of Logistics and Supply Chain Management, Universidad Popular Autonóma del Estado de Puebla, Puebla, Puebla, Mexico

## ABSTRACT

The Capacitated Centered Clustering Problem (CCCP)—a multi-facility location model—is very important within the logistics and supply chain management fields due to its impact on industrial transportation and distribution. However, solving the CCCP is a challenging task due to its computational complexity. In this work, a strategy based on Gaussian mixture models (GMMs) and dispersion reduction is presented to obtain the most likely locations of facilities for sets of client points considering their distribution patterns. Experiments performed on large CCCP instances, and considering updated best-known solutions, led to estimate the performance of the GMMs approach, termed as Dispersion Reduction GMMs, with a mean error gap smaller than 2.6%. This result is more competitive when compared to Variable Neighborhood Search, Simulated Annealing, Genetic Algorithm and CKMeans and faster to achieve when compared to the best-known solutions obtained by Tabu-Search and Clustering Search.

# INTRODUCTION

Facilities are very important infrastructure within the supply chain as they support production, distribution and warehousing. Due to this, many operative processes associated to facilities are subject to optimization. Fields such as facility layout planning are crucial for smooth material handling and production flow (*Mohamadghasemi & Hadi-Vencheh, 2012*, *Hadi-Vencheh & Mohamadghasemi, 2013*; *Niroomand et al., 2015*).

On the other hand, where to locate facilities within specific regions is a central problem for strategic decisions of transportation and distribution (*Chaves, Correa & Lorena, 2007*). This is because the distance between the facilities and the customers (demand or client points) is crucial to provide efficient transportation and distribution services.

Within this context, the Capacitated Centered Clustering Problem (CCCP) is one of the best-known and most challenging multi-facility location (MFL) problems in the fields of operations research, logistics and supply chain management (*Chaves & Nogueira-Lorena, 2010*, *Mahmoodi-Darani et al., 2013*). The CCCP is focused on determining clusters or

Corresponding author
Santiago-Omar Caballero-Morales, santiagoomar.caballero@upaep.mx

groups of demand or client points which can lead to minimum average distances to their centroids (where facilities are to be located). The cumulative demand of the client points assigned to a cluster cannot exceed the capacity of the facility located at its centroid (*Chaves & Nogueira-Lorena, 2010*; *Chaves & Nogueira-Lorena, 2011*; *Negreiros & Palhano, 2006*).

The mathematical formulation of the CCCP is presented as follows (*Chaves & Nogueira-Lorena, 2011*; *Negreiros & Palhano, 2006*):

$$\min \sum_{i \in I} \sum_{k \in K} \|x_i - m_k\|^2 y_{ik} \tag{1}$$

subject to

$$\sum_{k \in K} y_{ik} = 1 \quad \forall i \in I \tag{2}$$

$$\sum_{i \in I} y_{ik} = n_k \quad \forall k \in K \tag{3}$$

$$\sum_{i \in I} x_i y_{ik} \leq n_k m_k \quad \forall k \in K \tag{4}$$

$$\sum_{i \in I} d_i y_{ik} \leq C_k \quad \forall k \in K. \tag{5}$$

$$m_k \in \Re^l, \ n_k \in N, \ y_{ik} \in \{0, 1\} \ \forall i \in I, \ \forall k \in K \tag{6}$$

where (a) $I$ is the set of all demand points or clients ($N$ = number of points); (b) $K$ is the set of clusters (group of clients assigned to a facility) with $|K|$ = number of facilities; (c) $x_i$ is the geometric position of the $i$-th point in the $\Re^l$ space ($l = 2$ for a two-dimensional space and $x_i = [cx_i \ cy_i]$ where $cx_i$ is the $x$-coordinate and $cy_i$ is the $y$-coordinate of $x_i$); (d) $m_k$ is the geometric position of the centroid of a cluster $k$ (i.e., the location of the $k$-th facility, and if $l = 2$, $m_k = [cx_k \ cy_k]$ where $cx_k$ is the $x$-coordinate and $cy_k$ is the $y$-coordinate of $m_k$); (e) $y_{ik} = 1$ if the point $i$ is assigned to cluster $k$ and $y_{ik} = 0$ otherwise; (f) $n_k$ is the number of points in the cluster $k$; (g) $d_i$ is the demand of the point $i$; and (h) $C_k$ is the capacity of each cluster $k$.

In this formulation Eq. (1) defines the non-linear objective function which consists on minimizing the total distance between each point and the centroid of the cluster where the point is assigned. As mentioned in *Chaves & Nogueira-Lorena (2010)* the geometric position of the centroid depends on the points that compose the cluster. Equations (2) and (3) are restrictions that define that each point is only assigned to one cluster and provides the number of points in each cluster respectively. Equation (4) is the restriction that locates the centroid of each cluster at its geometric center while Eq. (5) is the restriction that defines that the total demands of the points assigned to a cluster $k$ must not exceed its capacity. Finally, Eq. (6) define the nature of $m_k$, the decision variable $y_{ik}$ and the upper limits for the number of individuals or points per cluster ($n_k$). $n_k$ is obtained from the values of the decision variables $y_{ik}$ as these determine the number of points to be assigned at each cluster (i.e., if for $k = 1$ the optimal solution is $y_{20,1} = y_{13,1} = y_{46,1} = 1$, then $n_1 = 3$). This is explained in more detail in Section "Standard EM Algorithm".

The CCCP has been approached with different solving methods (particularly meta-heuristics) due to its combinatorial nature and NP-hard computational complexity (hence, it is difficult to solve it with exact methods) (*Chaves, Correa & Lorena, 2007*; *Chaves & Nogueira-Lorena, 2010*; *Herda, 2015*; *Negreiros & Palhano, 2006*). While there are works reported in the literature that have achieved very competitive results for small, medium and large instances of the CCCP, their performance is dependent of the size or scale of the instances.

In this work the technique of Gaussian mixture models (GMMs) is proposed to estimate clusters of maximum probability in order to provide feasible solutions for large instances of the CCCP ($N > 1,000$ points). This is accomplished by the meta-heuristic termed as Dispersion Reduction GMMs (DRG) which integrates the following algorithms:

- an adapted Expectation-Maximization (EM) algorithm to estimate the parameters of GMMs and generate clusters of points of maximum likelihood with the capacity requirements of the CCCP;
- a dispersion reduction algorithm to compress large CCCP data and achieve near optimal results within faster computational times.

Both, the application of the GMMs and the validity of the EM algorithm for the CCCP, have not been explored in previous works. Experiments of the DRG on large instances of the CCCP considering the benchmarks of the Clustering Search (CS) algorithm (*De-Oliveira, Chaves & Nogueira-Lorena, 2013*) and Tabu-Search (TS) (*Fernandes-Muritiba et al., 2012*) led to a mean error less than 2.6%. In general, the DRG algorithm performed better than Variable Neighborhood Search (VNS), Simulated Annealing (SA), Genetic Algorithms (GA) and CKMeans (CKM). The advances of the present work can provide a better understanding of Gaussian modeling and the EM algorithm for their application in facility location problems.

The contributions of this paper are presented as follows: First, in Section "Clustering" an overview of clustering and the mathematical foundations of GMMs are presented. Also, the findings regarding the specific dispersion reduction process required by CCCP data in order to make it suitable for the EM algorithm are presented and discussed. Then, in Section "DRG Meta-heuristic" the details of the DRG meta-heuristic are presented while the results regarding its performance are discussed in Section "Results and Assessment". Finally, conclusions and improvement recommendations are discussed in Section "Conclusions and Future Work".

## CLUSTERING

Most of the solving algorithms for the CCCP perform clustering within the search process for initial partitioning of the set of demand points. This partitioning is then improved by exchanging certain points between the most promising partitions or clusters (*Hansen & Jaumard, 1997*; *Negreiros & Palhano, 2006*).

Formally, clustering involves the grouping of data points in a way that homogeneity of points within a group, and the heterogeneity of points between groups, are maximized (*Chaves & Nogueira-Lorena, 2010*; *Negreiros & Palhano, 2006*). For the CCCP the technique used for clustering has an important role in the quality of the solutions, which

may lead to significant differences from best-known results (i.e., errors from to 15.0% (*Radiah, Hasnah & Omar, 2013*) to 50% (*Negreiros & Palhano, 2006*)).

## Gaussian mixture models

Gaussian Mixture Models (GMMs) have been widely studied for data modeling within the field of pattern classification based on Bayes decision theory (*Theodoridis & Koutroumbas, 2010*). For more accurate modeling of multi-dimensional patterns a mixture of Gaussian distributions can be used. Each mixture component is defined by two main parameters: a mean vector and a covariance matrix (*Forsyth, 2012*; *Theodoridis & Koutroumbas, 1979*; *Theodoridis & Koutroumbas, 2010*). Within this context, "clusters" can be characterized by each Gaussian component (mixture) which can model sub-sets of the whole set of patterns with maximum likelihood.

There are important differences between clustering that can be obtained with GMMs and centroid-based clustering techniques (such as K-Means or K-Nearest Neighbors). The following can be mentioned:

- Over-fitting (e.g., the model "overreacts" to minor fluctuations in the training data for prediction purposes) can be avoided with Gaussian distribution-based clustering.
- Clusters defined with Gaussian distributions can have different shapes and sizes. By contrast, centroid-based clustering algorithms tend to find clusters of comparable size of (more or less) symmetrical shape (*Mohammed, Badruddin & Mohammed, 2016*).
- At each iteration, Gaussian distribution-based clustering performs, for a given point, a "soft-assignment" to a particular cluster (there is a degree of uncertainty regarding the assignment). The centroid-based clustering performs a hard-assignment (or direct assignment) where a given point is assigned to a particular cluster and there is no uncertainty.

Due to these differences, the GMM-based clustering was considered as an alternative to generate feasible solutions for the CCCP. In terms of the CCCP formulation described in Section "Introduction" a cluster can be modeled by a single Gaussian probability density function (PDF). Hence, the location "patterns" of a set of clients $X$ can be modeled by a mixture of $K$ Gaussian PDFs where each PDF models a single cluster. If the set contains $N$ clients, then $X = [x_{i=1}, x_{i=2}, \ldots, x_{i=N}]$ and the mixture can be expressed as:

$$p(X) = \sum_{k=1}^{K} P_k p(X|k) = \sum_{k=1}^{K} P_k \mathcal{N}(X|m_k, S_k), \tag{7}$$

where $k = 1, \ldots, K$ and $|K| = p$ is the number of Gaussian PDFs, $p(X \mid k)$ represents the probabilities of each Gaussian PDF describing the set of clients $X$ (*Theodoridis & Koutroumbas, 2010*) and $P_k$ is the weight associated to each Gaussian PDF (hence, $\sum_{k=1}^{K} P_k = 1.0$). Each Gaussian component can be expressed as:

$$p(X|k) = \mathcal{N}(X|m_k, S_k) = \frac{1}{\sqrt{2\pi|S_k|}} e^{-\frac{1}{2}(X - m_k)^T S_k^{-1}(X - m_k)} \quad \forall k \in K, \tag{8}$$

where $m_k$ is the mean vector and $S_k$ is the covariance matrix of the $k$-th Gaussian PDF or $k$-th cluster. Note that $m_k$ and each element of $X$ (i.e., any $x_i$) must have the same size or dimension which is defined by $l$ (in this case, $l = 2$ because each point consists of a $x$-coordinate and a $y$-coordinate). Finally, $S_k$ is a matrix of size $l \times l$.

For clustering purposes, $m_k$ can model the mean vector of a sub-set of points in $X$ which is more likely to be described by the Gaussian PDF $k$ as estimated by Eq. (8). If the points in this sub-set of $X$ are clustered, then $m_k$ can represent the "centroid" of the cluster $k$ and $S_k$ can model the size and shape of the cluster $k$ (*Bishop, 2006*; *Theodoridis & Koutroumbas, 2010*). Observe that $X - m_k$ defines a distance or difference between each point in $X$ and the centroid (located at $m_k$) of each cluster $k$. Thus, the estimation of probabilities considers the distance between each point $x_i$ and each cluster $k$.

The parameters of the Gaussian PDFs ($P_k$, $m_k$ and $S_k$) that best model (describe) each sub-set of the whole pattern $X$ can be estimated by the iterative algorithm of Expectation-Maximization (EM) (*Bishop, 2006*; *Theodoridis & Koutroumbas, 2010*).

The advantage of this Gaussian approach for clustering is that faster inference about the points $x_i$ that belong to a specific cluster $k$ may be obtained considering all points. In this context, it is important to mention that due to the probabilistic nature of the inference process, a single point $x_i$ is not directly assigned to a specific cluster (as it is required by the CCCP) because all points have probabilities associated to all clusters (i.e., "soft-assignment"). Also, this approach does not consider the capacity of each cluster and the demand associated to each client point. Thus, restrictions about the quantity of points $x_i$ associated to each cluster are not integrated in this algorithm. In order to consider these requirements and restrictions a modification on the EM algorithm was performed. This is described in the following sections.

## Dispersion reduction in GMMs performance for the CCCP

Capacitated Centered Clustering Problem data, which consists of $x$ and $y$ coordinates, represents a particular challenge for GMMs. This is because the values of coordinates can affect the computation of the exponential element of Eq. (8). Also, dispersion of data may affect convergence of the EM algorithm for $P_k$, $m_k$ and $S_k$. Thus, specific re-scaling or coding of CCCP data was required to reduce the effect of dispersion and coordinates' values on the computations of GMMs.

In order to reduce dispersion of the CCCP data the compression algorithm presented in Fig. 1 was performed. It is important to mention that compression is only performed on the coordinates' values, and not on the number of data points (thus, the size of the instance remains the same).

As presented, the original $x$–$y$ coordinates were replaced by their unique indexes. This led to elimination of empty spaces between data points. Coding of the compressed data was performed within the interval $[0, 1]$ as presented in Fig. 1. This coding made the compressed data more suitable for fast computation of Eq. (8).

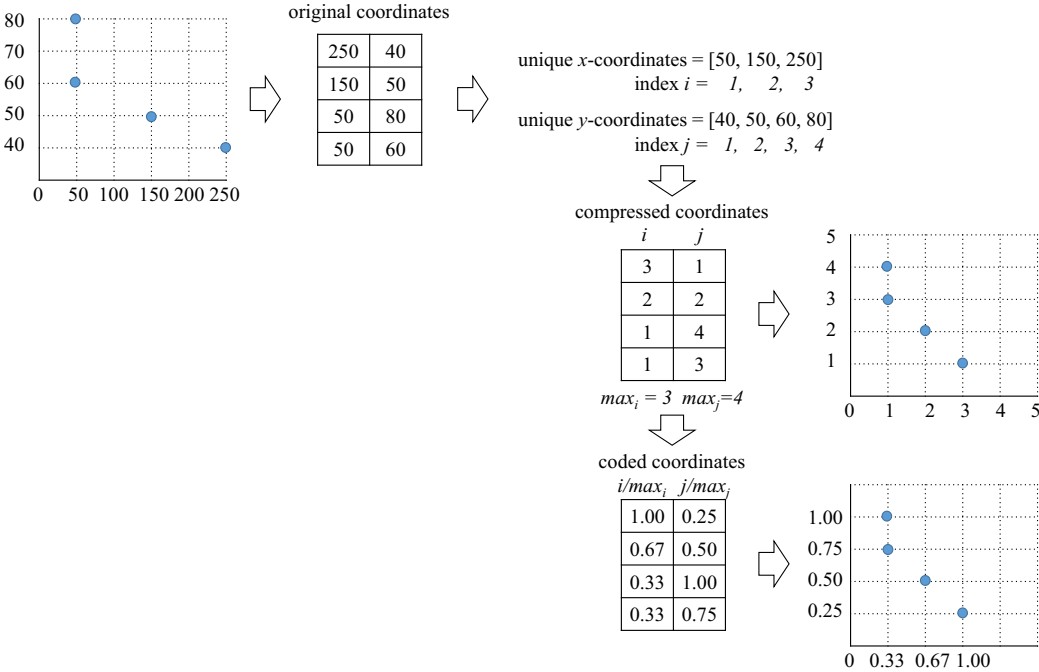

**Figure 1 CCCP data compression and coding algorithm (dispersion reduction algorithm).**

An important issue regarding the use of GMMs for the CCCP is the restriction on the number of Gaussian components. In this regard, extensive research has been performed to determine the most suitable number of Gaussian components for different sets of data (*McLachlan & Rathnayake, 2014*). However, available research does not consider the capacity aspect of clustering which is the main feature of CCCP data. Nevertheless, a restriction on the number of Gaussian components must be considered because, as discussed in *Fraley & Raftery (1998)*, the EM algorithm may not be suitable for cases with very large numbers of components.

Due to this situation a maximum number of 30 Gaussian components was considered. This restriction was based on the *DONI* database, which has some of the CCCP instances with the largest number of client points (*Fernandes-Muritiba et al., 2012*) (i.e., $N = 13,000$ client points with $p = 30$ facilities). A discussion on future extensions for the Gaussian approach and the EM algorithm to address larger number of components is presented in the final section.

## DRG META-HEURISTIC

### Standard EM algorithm

Figure 2 presents the structure of the standard EM algorithm (*Bishop, 2006*; *Theodoridis & Koutroumbas, 1979*; *Theodoridis & Koutroumbas, 2010*) considering the variables defined by Eqs. (7) and (8). In this case, $X$ represents the array of compressed and coded $x - y$ coordinates of all points $x_i$ (the array structures presented in Figs. 2 and 3 follow the EM formulation described in *Theodoridis & Koutroumbas (1979)*).
**CCCP Instance Data**

$X$ = matrix of dimension $l \times N$ that contains the $x$-coordinate ($cx_i$) and the $y$-coordinate ($cy_i$) of each point $x_i$

$$[\ x_1\ ,\ \ x_2\ ,\ \ x_3\ ,\ \ x_4\ ,\ \ \dots\ ,\ \ x_N\ ]$$

| | $cx_1$ | $cx_2$ | $cx_3$ | $cx_4$ | ... | $cx_N$ |
|---|---|---|---|---|---|---|
| $X =$ | $cy_1$ | $cy_2$ | $cy_3$ | $cy_4$ | ... | $cy_N$ |

$l = 2$ ($x$-coordinate and $y$-coordinate of each point $x_i$)

1.  Initialization: Initialize parameters of the Gaussian PDFs: $m_k$, $S_k$, and $P_k$.

2.  Expectation: Assign to each point $x_i$ an assignment score $\gamma(z_{ik})$ for each cluster $k$. This score represents the responsibility of the Gaussian PDF $k$ on the generation of the point $x_i$.

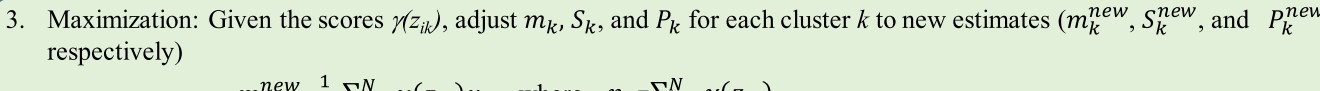

$$\gamma(z_{ik}) = \frac{P_k \mathcal{N}(x_i | m_k, S_k)}{\sum_{j=1}^{K} P_j \mathcal{N}(x_i | m_j, S_j)}$$

3.  Maximization: Given the scores $\gamma(z_{ik})$, adjust $m_k$, $S_k$, and $P_k$ for each cluster $k$ to new estimates ($m_k^{new}$, $S_k^{new}$, and $P_k^{new}$ respectively)

$$m_k^{new} = \frac{1}{n_k} \sum_{i=1}^{N} \gamma(z_{ik}) x_i \quad \text{where} \quad n_k = \sum_{i=1}^{N} \gamma(z_{ik})$$

$$S_k^{new} = \frac{1}{n_k} \sum_{i=1}^{N} \gamma(z_{ik}) (x_i - m_k^{new})(x_i - m_k^{new})^T$$

$$P_k^{new} = \frac{n_k}{N}$$

4.  Evaluation: Stop if parameters converge. Otherwise, go to Step 2.
    Convergence is achieved if:

$$\sum_{k=1}^{K} [|m_k - m_k^{new}| + |S_k - S_k^{new}| + |P_k - P_k^{new}|] < e$$

otherwise: $m_k \leftarrow m_k^{new} \quad S_k \leftarrow S_k^{new} \quad P_k \leftarrow P_k^{new}$

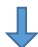

**Figure 2  Structure of the standard expectation-maximization (EM) algorithm.**

As presented in Fig. 2, the locations of the $N$ clients are stored into the array $X$ which consists of a matrix of dimension $l \times N$ where: (a) $N$ is the number of client points, and (b) $l = 2$ as each column vector of $X$ consists of the values $cx_i$ and $cy_i$ that identify the compressed and coded $x - y$ coordinates of the client point $x_i$ where $i = 1, \dots, N$.

The EM algorithm starts with initial values for $m_k$, $S_k$ and $P_k$. Values for $m_k$ and $S_k$ were randomly generated as follows:

$$m_k = \left[ \text{random}(cx_{min}, cx_{max}), \text{random}(cy_{min}, cy_{max}) \right]^T \forall k \in K, \tag{9}$$

$$S_k = \text{random}(0.0, 0.1) \times I_l \ \forall k \in K, \tag{10}$$

where $(cx_{min}, cx_{max})$ and $(cy_{min}, cy_{max})$ are the minimum and maximum values throughout all compressed and coded $x$ and $y$ coordinates respectively, and $I_l$ is the identity matrix of size $l \times l$.

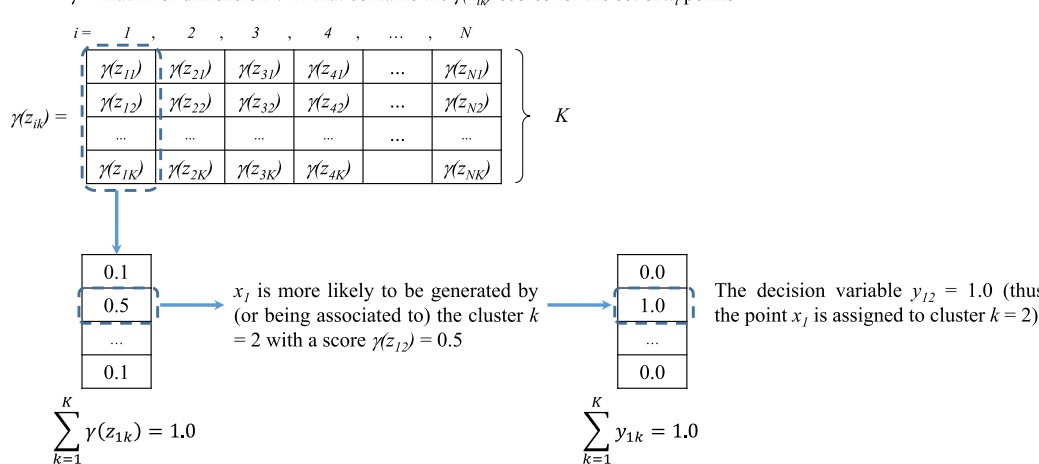

**Figure 3  Assignment of values for $y_{ik}$ from the scores of $\gamma(z_{ik})$.**

For $P_k$ a lower bound for $K$ was obtained by considering the total demand of the points $x_i$ and the capacity of the clusters $C_k$. Because all clusters have the same capacity, $C_k = C$. Then, $K$ and $P_k$ were obtained as follows:

$$K = \frac{\sum_{i=1}^{N} d_i}{C}, \tag{11}$$

$$P_k = \frac{1}{|K|} \quad \forall k \in K. \tag{12}$$

The stage of Expectation starts with these initial values for $m_k$, $S_k$ and $P_k$. An initial computation of assignment or "responsibility" scores $\gamma(z_{ik})$ is performed to determine which $x_i$ is more likely to be associated to a particular cluster (and thus, to belong to this cluster) with parameters $m_k$, $S_k$ and $P_k$ (*Bishop, 2006*). Observe that, as presented in Fig. 2 (Step 2), $\gamma(z_{ik})$ is computed by means of Eq. (8). These scores can lead to provide values for the decision variable $y_{ik}$ of the CCCP objective function (see Eq. (1)). This process will be discussed in the following section.

Then, the stage of Maximization integrates the scores $\gamma(z_{ik})$ into the re-estimation of the cluster's parameters which leads to $m_k^{new}$, $S_k^{new}$ and $P_k^{new}$. Convergence is achieved if the total error or absolute difference between the previous and new estimates is less than a given threshold $e$ (in this case, $e=0.5$). If convergence is not achieved, then $m_k \leftarrow m_k^{new}$, $S_k \leftarrow S_k^{new}$ and $P_k \leftarrow P_k^{new}$.

## Modified EM algorithm with direct assignment and capacity restrictions

As discussed in Section "Gaussian Mixture Models" the advantage of the GMMs approach is that faster inference about the points $x_i$ that belong to a specific cluster $k$ can be obtained. This inference is performed based on the probabilities defined by Eq. (8) where the

parameters of the Gaussian PDFs are estimated with the standard EM algorithm. However under this approach a single point $x_i$ is not directly assigned to a specific cluster (as it is required by the CCCP) because all points have probabilities associated to all clusters (i.e., "soft-assignment"). Also, while the standard EM algorithm can determine the most likely sub-sets of client points in $X$ to be covered by each cluster $k$, these sub-sets are not restricted by the capacity of the cluster and the demand of each assigned client point. Hence, the standard EM algorithm was modified in order to comply with the requirements and restrictions of the CCCP.

As previously mentioned, the score $\gamma(z_{ik})$ computed in Step 2 of the EM algorithm (see Fig. 2) represents the likelihood or responsibility of the cluster $k$ on the generation of the point $x_i$ (*Bishop, 2006*). As presented in Fig. 3 all scores are stored into a matrix $\gamma$ of dimension $K \times N$, where each column vector $[\gamma(z_{i1}), \ldots, \gamma(z_{iK})]^T$ contains the assignment scores of the point $x_i$ to all clusters (thus, $\sum_{k=1}^{K} \gamma(z_{ik}) = 1.0 \ \forall i \in N$). These scores represent the basis for defining the decision variable $y_{ik}$.

From Section "Introduction" it was explained that $y_{ik} = 1$ if the point $x_i$ is assigned to cluster $k$ and $y_{ik} = 0$ otherwise (each point $x_i$ can be assigned to only one cluster). Based on the scores of $\gamma(z_{ik})$ it was determined that for each vector $x_i$ the cluster assignment would be defined by the cluster $k$ with maximum $\gamma(z_{ik})$ value. If two or more clusters share the same maximum likelihood, then one of them is randomly assigned. An example of this assignment process is presented in Fig. 3.

By determining the unique assignment of each point $x_i$ to each cluster $k$ at Step 2 of the EM algorithm (see Fig. 2), the number of points assigned to each cluster is obtained. This leads to determine the cumulative demand of the points assigned to each cluster. This information is stored into the vector:

$$\text{Demands} = [D_1, D_2, \ldots, D_K], \tag{13}$$

where $D_k$ represents the cumulative demand of the points assigned to cluster $k$ and it must satisfy $D_k \leq C_k$. This vector is important to comply with the capacity restrictions because it was found that homogenization of the cumulative demands $D_k$ contributes to this objective. Homogenization is achieved by minimizing the coefficient of variation between all cumulative demands:

$$\min \text{CV} = \frac{\text{Standard Deviation(Demands)}}{\text{Mean(Demands)}}. \tag{14}$$

The objective function defined by Eq. (14) is integrated within the evaluation step of the standard EM algorithm (see Step 4 of Fig. 2). This leads to the modified EM algorithm with capacity restrictions where convergence is based on two objectives:

- minimization of the error (less than a threshold of $e = 0.5$) between the estimates of $m_k$, $P_k$ and $S_k$;
- minimization of the coefficient of variation of *Demands*.

An important issue regarding the minimization of CV is that there is the possibility that min CV may not lead to comply the condition $D_k \leq C_k$ for all clusters $k$. This was addressed

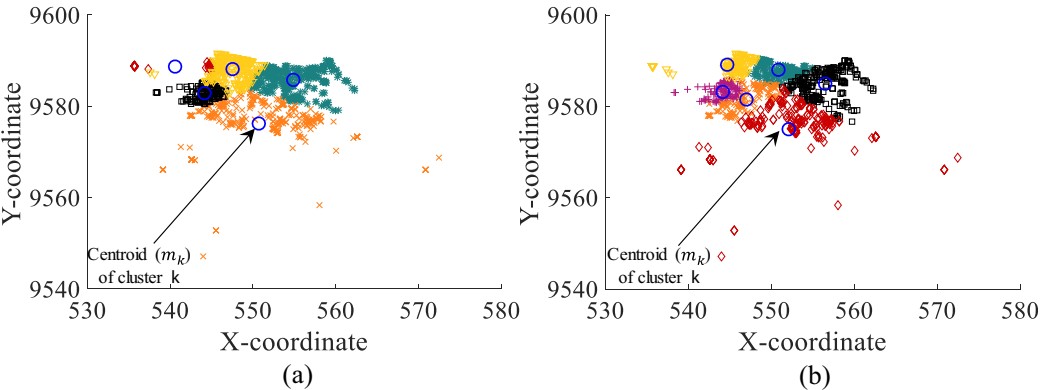

**Figure 4 Clustering with (A) standard EM algorithm and (B) DRG meta-heuristic (modified EM algorithm with decoded and uncompressed CCCP data).**

by the following strategy: if min CV leads to a cluster $k$ where the condition $D_k \leq C_k$ is not complied, then the farthest assigned points are re-assigned to other clusters with closer centroids and available capacity.

Figure 4 presents an example of the clustering achieved without and with the objective function defined by Eq. (14). As presented, the clusters are more accurately defined with the integration of Eq. (14). Also, clusters comply with the capacity restrictions of the CCCP.

It is important to remember that the proposed DRG meta-heuristic consists of the modified EM algorithm which provides the cluster assignation for each point $x_i$ considering compressed and coded CCCP data. Thus, Fig. 4 presents the decoded and uncompressed points $x_i$ (i.e., original $cx_i$ and $cy_i$ coordinates) assigned to each cluster based on the assignments of the DRG meta-heuristic with the modified EM algorithm on compressed and coded data.

Further improvement of the assignments and the centroids at $m_k$ are achieved by a Greedy algorithm that verifies that all points $x_i$ are assigned to the closest cluster. This leads to additional exchange and insertion/deletion of complying client points $x_i$ between clusters. These operations must comply with the capacity restrictions of each cluster. The assignments are updated if the insertion/deletion is valid. This leads to re-estimation of the locations of the $m_k$ centroids.

## RESULTS AND ASSESSMENT

Implementation of the DRGMMs meta-heuristic was performed with the software *MATLAB* on a laptop computer with the following hardware: Intel CORE i7-4500U CPU at 2.40 GHz with 8 GB RAM. For testing purposes the libraries presented in Table 1 were considered.

For comparison purposes the methods and best-known results presented in *Chaves & Nogueira-Lorena (2010)*, *Chaves & Nogueira-Lorena (2011)*, *Fernandes-Muritiba et al. (2012)*, *De-Oliveira, Chaves & Nogueira-Lorena (2013)*, *Negreiros & Palhano (2006)*,

**Table 1 Libraries of CCCP instances considered for testing and assessment.**

| Instance | N | K |
|----------|------|----|
| doni1 | 1,000 | 6 |
| doni2 | 2,000 | 6 |
| doni3 | 3,000 | 8 |
| doni4 | 4,000 | 10 |
| doni5 | 5,000 | 12 |
| doni6 | 10,000 | 23 |
| doni7 | 13,221 | 30 |
| SJC1 | 100 | 10 |
| SJC2 | 200 | 15 |
| SJC3a | 300 | 25 |
| SJC4a | 402 | 30 |
| TA25 | 25 | 5 |
| TA50 | 50 | 5 |
| TA60 | 60 | 5 |
| TA70 | 70 | 5 |
| TA80 | 80 | 7 |
| TA90 | 90 | 4 |
| TA100 | 100 | 6 |

*Palhano, Negreiros & Laporte (2008)* and *Pereira & Senne (2008)* for the CCCP were considered. These methods are the following:

- Best-known solutions*. The benchmarks reported in *De-Oliveira, Chaves & Nogueira-Lorena (2013)* have been used for the assessment of the most efficient methods for the CCCP. For this work, these benchmarks were updated with the best-known results obtained with TS (*Fernandes-Muritiba et al., 2012*) which is considered as the current state of the art algorithm for the CCCP (*Carvalho, Mendes & Azeredo-Chaves, 2017*; *Pereira & Carvalho, 2017*).
- Clustering Search (CS). Hybrid method which combines meta-heuristics and local search heuristics in which the search is intensified only in areas of the search space that deserve special attention (promising regions). The main idea of CS is to identify promising areas of the search space by generating solutions through a meta-heuristic and "clustering" them into groups that are further explored with local search heuristics (*Chaves & Nogueira-Lorena, 2010*; *Chaves & Nogueira-Lorena, 2011*; *De-Oliveira, Chaves & Nogueira-Lorena, 2013*). The results reported in *De-Oliveira, Chaves & Nogueira-Lorena (2013)* were considered for comparison with DRG.
- A two-phase heuristic using VNS. The results reported in *Negreiros & Palhano (2006)* were considered for comparison with DRG. These results were also reviewed by *Fernandes-Muritiba et al. (2012)* for comparison with TS.

- CKMeans (CKM). The results reported in *Palhano, Negreiros & Laporte (2008)* were considered for comparison with DRG. These results were also reviewed by *Chaves & Nogueira-Lorena (2011)* for comparison with GA.

- Two-Column Generation (TCG). The results reported in *Pereira & Senne (2008)* were considered for comparison with DRG.

- Simulated Annealing (SA) and Genetic Algorithm (GA). CS performs a clustering strategy on solutions generated by a meta-heuristic, and the research in *Chaves & Nogueira-Lorena (2010, 2011)* reported the comparison of results of the CS method with the SA and GA meta-heuristics respectively. For assessment purposes, (*Chaves & Nogueira-Lorena, 2010, 2011*) also reported the comparison of results of SA and GA without the CS strategy (e.g., independent performance of SA and GA). These independent results were considered for comparison with DRG.

- Tabu-Search (TS). The TS algorithm reported in *Fernandes-Muritiba et al. (2012)* is currently considered state of the art by Rodrigo de Carvalho et al. for the CCCP (*Carvalho, Mendes & Azeredo-Chaves, 2017*; *Pereira & Carvalho, 2017*). The results reported in *Fernandes-Muritiba et al. (2012)* were considered for comparison with DRG.

In order to compute the error, gap or deviation from the updated best-known solutions the error metric presented by *Yousefikhoshbakht & Khorram (2012)* was considered:

$$\text{Error}(\%) = 100 \times \left[\frac{a - b}{b}\right], \tag{15}$$

where $a$ is the cost or distance of the best solution found by the algorithm for a given instance while $b$ is the best known solution for the same instance. In this case it is important to mention that the best-known solution is not necessarily the optimal solution due to the NP-hard complexity of the CCCP. Initially, this metric was computed for the DRG, VNS, SA, CS, TS and GA methods because the reference data was available for all sets of instances.

## DRG vs. VNS-SA-CS-TS-GA

Table 2 presents the best results of the DRG meta-heuristic for the considered instances. Information regarding the runs performed by each method to report the best result is also presented when available. Also, information regarding the programing language and the hardware used by the authors of the reviewed methods were also included.

As reported in the literature, CS and TS are the most competitive methods to solve the CCCP with a mean error gap of 0.78% and 0.61% respectively (and thus, were considered to update the benchmark solutions). Then, the proposed DRG meta-heuristic stands as the next most competitive method with a mean error gap of 2.58%. The VNS, SA and GA methods show a more significant mean error gap with 4.69%, 13.79% and 7.03% respectively. Also, VNS, SA and GA show a higher variability in performance which is characterized by an increased standard deviation when compared with their mean error gap. The DRG shows a balanced mean and standard deviation, thus its performance is more robust and consistent through all different CCCP instances.

**Table 2** Performance of DRG, VNS, SA, CS, TS and GA on CCCP instances when compared to updated Best-known solutions*: (a) MATLAB, Intel CORE i7 at 2.4 GHz and 8 GB RAM, (b) AMD ATHLON at 1.6 GHz and 512 MB RAM, (c) C++, Pentium 4 at 3.0 GHz, (d) C++, Pentium 4 at 3.0 GHz, (e) C++, Intel CORE 2 Quad Q9550 CPU at 2.83 GHz and 4 GB RAM, (f) C++, Pentium 4 at 3.0 GHz.

| Instance | Best known* | N | K | DRG (a) (10 runs) | Error (%) | VNS (b) | Error (%) | SA (c) (10 runs) | Error (%) | CS (d) (20 runs) | Error (%) | TS (e) (25 runs) | Error (%) | GA (f) (20 runs) | Error (%) |
|---|---|---|---|---|---|---|---|---|---|---|---|---|---|---|---|
| TA25 | 1,251.44 | 25 | 5 | 1,256.62 | 0.41 | 1,251.44 | 0.00 | 1,273.46 | 1.76 | 1,251.44 | 0.00 | 1,251.45 | 0.00 | 1,273.46 | 1.76 |
| TA50 | 4,474.52 | 50 | 5 | 4,476.92 | 0.05 | 4,476.12 | 0.04 | 4,478.15 | 0.08 | 4,474.52 | 0.00 | 4,474.52 | 0.00 | 4,474.52 | 0.00 |
| TA60 | 5,356.58 | 60 | 5 | 5,356.58 | 0.00 | 5,356.58 | 0.00 | 5,370.05 | 0.25 | 5,356.58 | 0.00 | 5,356.58 | 0.00 | 5,356.58 | 0.00 |
| TA70 | 6,240.67 | 70 | 5 | 6,270.45 | 0.48 | 6,241.55 | 0.01 | 6,267.89 | 0.44 | 6,240.67 | 0.00 | 6,240.67 | 0.00 | 6,267.89 | 0.44 |
| TA80 | 5,515.46 | 80 | 7 | 5,748.30 | 4.22 | 5,730.28 | 3.89 | 5,780.55 | 4.81 | 5,730.28 | 3.89 | 5,730.28 | 3.89 | 5,775.69 | 4.72 |
| TA90 | 8,899.05 | 90 | 4 | 9,069.85 | 1.92 | 9,103.21 | 2.29 | 9,069.85 | 1.92 | 9,069.85 | 1.92 | 9,069.85 | 1.92 | 9,133.35 | 2.63 |
| TA100 | 8,102.04 | 100 | 6 | 8,122.36 | 0.25 | 8,122.67 | 0.25 | 8,153.64 | 0.64 | 8,102.04 | 0.00 | 8,102.04 | 0.00 | 8,189.44 | 1.08 |
| SJC1 | 17,359.75 | 100 | 10 | 17,492.77 | 0.77 | 17,696.53 | 1.94 | 17,363.47 | 0.02 | 17,359.75 | 0.00 | 17,359.75 | 0.00 | 17,363.47 | 0.02 |
| SJC2 | 33,181.65 | 200 | 15 | 33,317.03 | 0.41 | 33,423.84 | 0.73 | 33,458.40 | 0.83 | 33,181.65 | 0.00 | 33,181.65 | 0.00 | 33,324.04 | 0.43 |
| SJC3a | 45,354.38 | 300 | 25 | 46,395.96 | 2.30 | 47,985.29 | 5.80 | 46,847.61 | 3.29 | 45,354.38 | 0.00 | 45,356.35 | 0.00 | 46,682.60 | 2.93 |
| SJC4a | 61,931.60 | 402 | 30 | 62,701.95 | 1.24 | 66,689.96 | 7.68 | 64,981.66 | 4.92 | 61,931.60 | 0.00 | 61,993.66 | 0.10 | 65,978.89 | 6.54 |
| DONI1 | 3,021.41 | 1,000 | 6 | 3,074.97 | 1.77 | 3,021.41 | 0.00 | 3,138.67 | 3.88 | 3,022.26 | 0.03 | 3,025.12 | 0.12 | 3,122.02 | 3.33 |
| DONI2 | 6,080.70 | 2,000 | 6 | 6,456.14 | 6.17 | 6,080.70 | 0.00 | 6,985.30 | 14.88 | 6,372.81 | 4.80 | 6,384.84 | 5.00 | 6,394.96 | 5.17 |
| DONI3 | 8,343.49 | 3,000 | 8 | 8,911.87 | 6.81 | 8,769.05 | 5.10 | 9,653.27 | 15.70 | 8,438.96 | 1.14 | 8,343.49 | 0.00 | 8,945.88 | 7.22 |
| DONI4 | 10,777.64 | 4,000 | 10 | 11,453.68 | 6.27 | 11,516.14 | 6.85 | 13,328.16 | 23.66 | 10,854.48 | 0.71 | 10,777.64 | 0.00 | 11,130.16 | 3.27 |
| DONI5 | 11,114.67 | 5,000 | 12 | 11,776.59 | 5.96 | 11,635.18 | 4.68 | 13,920.49 | 25.24 | 11,134.94 | 0.18 | 11,114.67 | 0.00 | 11,341.52 | 2.04 |
| DONI6 | 15,610.46 | 10,000 | 23 | 16,362.38 | 4.82 | 18,443.50 | 18.15 | 29,102.49 | 86.43 | 15,722.67 | 0.72 | 15,610.46 | 0.00 | 19,226.96 | 23.17 |
| DONI7 | 18,484.13 | 13,221 | 30 | 18,974.23 | 2.65 | 23,478.79 | 27.02 | 29,484.66 | 59.51 | 18,596.74 | 0.61 | 18,484.13 | 0.00 | 29,915.77 | 61.85 |
| | | | | Mean= | 2.58 | Mean= | 4.69 | Mean= | 13.79 | Mean= | 0.78 | Mean= | 0.61 | Mean= | 7.03 |
| | | | | StdDev= | 2.45 | StdDev= | 7.18 | StdDev= | 23.43 | StdDev= | 1.41 | StdDev= | 1.48 | StdDev= | 14.68 |

### DRG vs. CKM-TCG

Tables 3 and 4 present the results of the DRG meta-heuristic for the instances where reference data of the CKM and TCG methods were available. As presented in Table 3, the DRG meta-heuristic is more competitive than the CKM method. Also, as previously observed, the DRG is more consistent.

When compared to the TCG method, this is more competitive than the DRG approach even though the error gaps are minimal (less than 1.5%).

### Error and speed vs. size of the instance

Figure 5 presents the graphical review of the error gaps of all methods based on the size of the test instance. TS and CS are located on the *x*-axis as they are the benchmark methods. It can be observed that, as the size of the instance increases, the error gap of SA, GA and VNS significantly increases. TCG presents very small error gaps with small instances (less than 1,000 points) and CKM reports error gaps comparable to SA for small-medium size instances (less than 5,000 points). In contrast, DRG performs consistently through all instances, decreasing its error gap as the instance grows from medium to large size (up to 13,0221 points).

**Table 3 Performance of DRG and CKM on CCCP instances when compared to Best-known solutions[*].**

| Instance | DRG | Error (%) | CKM | Error (%) |
|---|---|---|---|---|
| SJC1 | 1,7492.77 | 0.77 | 20,341.34 | 17.18 |
| SJC2 | 33,317.03 | 0.41 | 35,211.99 | 6.12 |
| SJC3a | 46,395.96 | 2.30 | 50,590.49 | 11.54 |
| SJC4a | 62,701.95 | 1.24 | 69,283.05 | 11.87 |
| DONI1 | 3,074.97 | 1.77 | 3,234.58 | 7.06 |
| DONI2 | 6,456.14 | 6.17 | 6,692.71 | 10.06 |
| DONI3 | 8,911.87 | 6.81 | 9,797.12 | 17.42 |
| DONI4 | 11,453.68 | 6.27 | 11,594.07 | 7.58 |
| DONI5 | 11,776.59 | 5.96 | 11,827.69 | 6.42 |
| | Mean= | 3.52 | Mean= | 10.58 |
| | StdDev= | 2.70 | StdDev= | 4.36 |

**Table 4 Performance of DRG and TCG on CCCP instances when compared to Best-known solutions[*]: (a) MATLAB, Intel CORE i7 at 2.4 GHz and 8 GB RAM, (b) C++, Intel CORE 2 Duo at 2 GHz with 2 GB RAM.**

| Instance | DRG (a) | Error (%) | TCG (b) | Error (%) |
|---|---|---|---|---|
| TA25 | 1,256.62 | 0.41 | 1,280.49 | 2.32 |
| TA50 | 4,476.92 | 0.05 | 4,474.52 | 0.00 |
| TA60 | 5,356.58 | 0.00 | 5,357.34 | 0.01 |
| TA70 | 6,270.45 | 0.48 | 6,240.67 | 0.00 |
| TA80 | 5,748.30 | 4.22 | 5,515.46 | 0.00 |
| TA90 | 9,069.85 | 1.92 | 8,899.05 | 0.00 |
| TA100 | 8,122.36 | 0.25 | 8,168.36 | 0.82 |
| SJC1 | 17,492.77 | 0.77 | 17,375.36 | 0.09 |
| SJC2 | 33,317.03 | 0.41 | 33,357.75 | 0.53 |
| SJC3a | 46,395.96 | 2.30 | 45,379.69 | 0.06 |
| SJC4a | 62,701.95 | 1.24 | 61,969.06 | 0.06 |
| | Mean= | 1.10 | Mean= | 0.35 |
| | StdDev= | 1.28 | StdDev= | 0.71 |

Regarding speed, Fig. 6 presents the computational (running) times reported by the reviewed methods. While TS and CS are the benchmark methods, these take over 25,000 s to reach the best-known solution for the largest instance. Note that for these methods, their computational times exponentially increase for instances larger than 6,000 points.

In contrast, SA is very consistent with a computational time of approximately 1,000 s through all instances. GA significantly increases for instances larger than 6,000 points (up to 7,000 seconds for the largest instance). However, these methods have the largest error gaps as reviewed in Fig. 5. The speed pattern of DRG is very similar to that of GA, however, as reviewed in Fig. 5, its error gap is the closest to the benchmark methods for instances larger than 6,000 points.

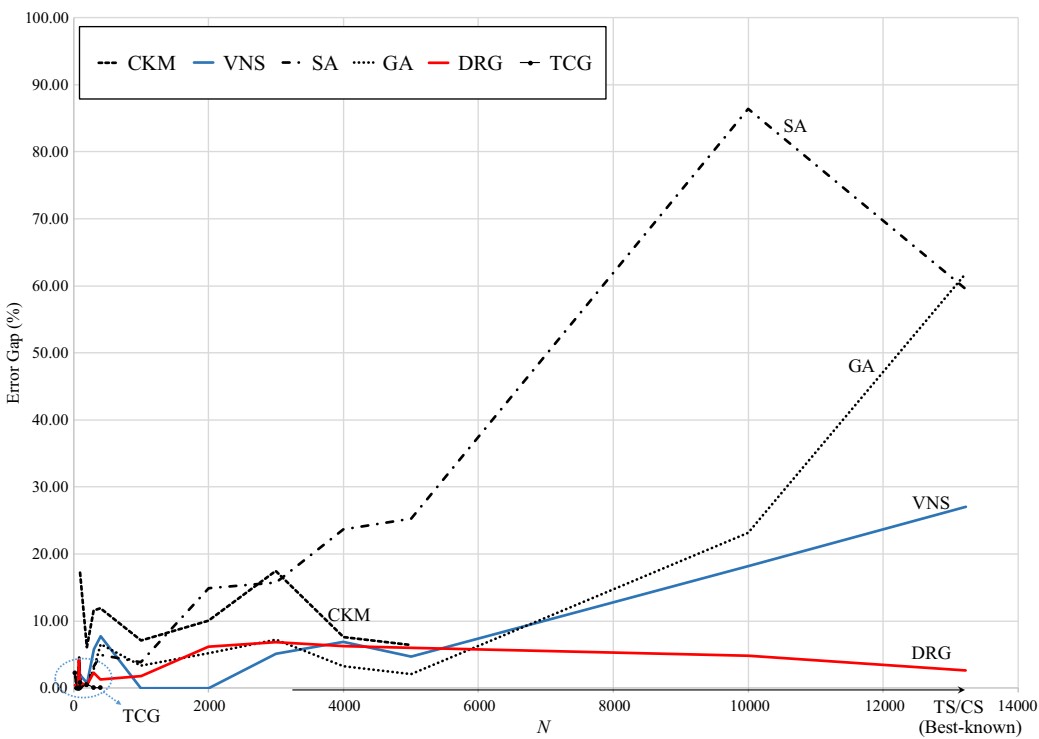

**Figure 5** Comparison of error gap vs. size of the instance.

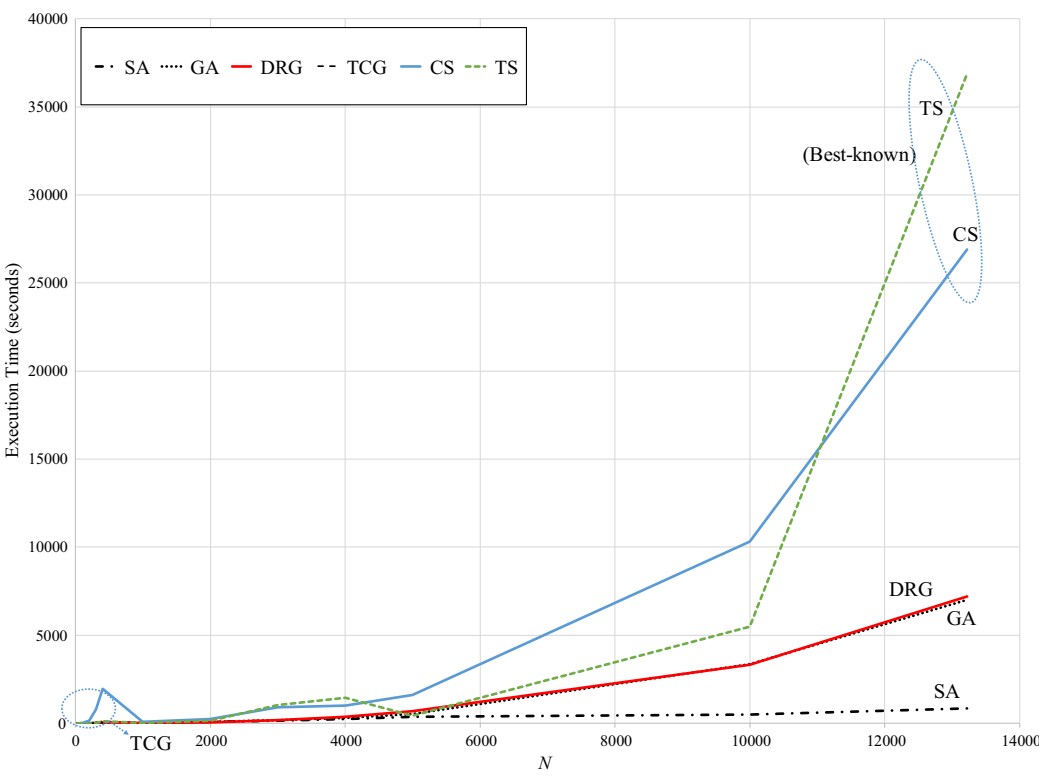

**Figure 6** Comparison of speed vs. size of the instance.

It is important to mention that this comparison may not be fair due to the differences in the programming language and the hardware used for implementation and testing of all the methods. In order to compare running speed all methods should be tested with the same hardware and be implemented with the same programming language by the same software developer (the developer's expertise may also affect the speed performance of the software). Due to the difficulty of achieving this task, in practice the comparison is commonly performed on the best results obtained by other methods as performed in *Chaves & Nogueira-Lorena (2011)* and *Fernandes-Muritiba et al. (2012)*. Running time is measured in order to determine if the proposed method can provide a solution within reasonable time considering standard resources of hardware and software. Due to this situation, the DRG can provide very suitable results (<2.6% error gap) within very reasonable time.

## CONCLUSIONS AND FUTURE WORK

Both, the application of Gaussian probability functions and the EM algorithm have not been explored in the literature as solving techniques for facility location problems (e.g., CCCP). An important aspect for the application of GMMs is the reduction of dispersion to accomplish more efficient clustering and convergence of the EM algorithm. Hence, the proposed DRG meta-heuristic provides important insights about the suitability of these techniques for the CCCP and the challenges to improve its performance.

Regarding performance, TS and CS are the most competitive solving methods for the CCCP and thus, were considered as benchmark methods with mean average error of 0.61% and 0.78% respectively. The proposed DRG meta-heuristic performed as the closest best method with a mean error gap smaller than 2.6% and there is evidence than it can provide these results faster when compared to TS and CS for large instances (>6,000 points). Hence, the DRG meta-heuristic can be a suitable alternative when compared to CS and TS regarding time, and a more efficient method when compared to VNS, SA, GA, and CKM.

The future work is focused on improving the performance of the DRG based on the following key aspects:

- The EM algorithm was found to be functional with up to 30 Gaussian components for the clustering process (see discussion on Section "Dispersion Reduction in GMMs Performance for the CCCP"). In this case the analysis of the Infinite Gaussian Mixture Model described in *Rasmussen (2010)* may lead to overcome this restriction.
- While most optimization methods such as those based on Mixed Integer Linear Programming (MILP) or meta-heuristics are purely quantitative, modeling of qualitative criteria may improve the optimization task. In example, in *Hadi-Vencheh & Mohamadghasemi (2013)* a methodology that integrated the Analytic Hierarchy Process (AHP) and a Nonlinear Programming Model (NLP) provided very suitable solution for a facility layout problem. Such approach may lead to improve the solving methods for other logistic problems such as the facility location problem.

- Integration with state-of-the-art meta-heuristics such as Migrating Birds Optimization (MBO) (*Niroomand et al., 2015*) and mat-heuristics (*Sartori & Buriol, 2020*).

### Funding
The author received no funding for this work.

### Competing Interests
The author declares that they have no competing interests.

### Author Contributions
- Santiago-Omar Caballero-Morales conceived and designed the experiments, performed the experiments, analyzed the data, performed the computation work, prepared figures and/or tables, authored or reviewed drafts of the paper, and approved the final draft.

### Data Availability
   Code and test instances of the considered databases are available in the Supplemental Files.

### Supplemental Information
Supplemental information for this article can be found online at http://dx.doi.org/10.7717/peerj-cs.332#supplemental-information.

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
