# Peer review of "Solution strategy based on Gaussian mixture models and dispersion reduction for the capacitated centered clustering problem"

_PeerJ Computer Science, doi:10.7717/peerj-cs.332_

## Round 0.1 · original submission · Major Revisions

The work is interesting. We encourage the authors to enhance the manuscript considering the major revisions suggested by reviewers.

Reviewer 1 ·

Basic reporting

This is a well-written and structured paper on the use of gaussian mixtures to address the issue of capacitated clustering. The problem is clearly stated and the method properly presented as an alternative way to solve this problem with less computational time

Experimental design

Expertiment were properly designed and sufficiently detailed, minor aspects of alternative testing instances are described.

Validity of the findings

Results appear to be sound. Maple code is clean and easy to understand.

Reviewer 2 ·

Basic reporting

see general comments

Experimental design

see general comments

Validity of the findings

see general comments

Additional comments

The paper proposes a new method to solve CCCP, called DRG. It is competitive against other methods in the literature.

The proposed method can beat all the non-benchmark methods in terms of accuracy. However, it seems for instance when N <= 10000 TS remains pretty competitive against DRG, so it is not quite clear to me under what circumstances one should consider DRG over other methods. For instance, what's the most common range of N in practical applications?

Also Figure 6 is a bit confusing because the legend says the red curve is DRG, while in the plot it is labeled as GMM.

Some typos in the paper:
All the section numbers seem to be missing such as in Line 81, 163, and 183.

Reviewer 3 ·

Basic reporting

This project is logically laid out and technically sound. It is also an enjoyable reading. The work should be completed as to demonstrate that the references are up to date (some performance benchmarks are almost ten years old).

Experimental design

Furthermore, albeit the performance of the proposed method is deemed novel and competitive, it fails to completely dominate other reported methods. Under what circumstances would an experimenter choose the proposed method over the other methods? This is particularly important considering that, in real life, the problem at hand is more strategic and does not quite critically requires the fastest possible heuristic solution (as it would be the case for an operational problem).

Validity of the findings

One important issue that must be discussed is the following: if the proposed method is meant to minimize spread, why doesn´t it completely dominate (in that criterion) the other methods that did not consider that objective explicitly?

Additionally, an appendix on graphical validation would help a lot to appreciate the kinds of solutions set forth by the proposed method.

Additional comments

In general, a probabilistic approach might be more complex than a deterministic approach. For this reason, more conclusive evidence of superior performance might be needed to accept the proposed method as an attractive competitor to other methods.

Reviewer 4 ·

Basic reporting

In this study the author proposed an integrated approach based on Gaussian mixture models and dispersion reduction to solve the capacitated centered clustering problem (CCCP). For this purpose, the author employed the proposed methodology to some CCCPs and the obtained results were compared with some methods in litarature. To my opinion, this study seems nice and interesting and can be accepted for publication. But, it needs some modification before its acceptance.

Experimental design

The experimental section is adequate and I think this part is good.

Validity of the findings

The findings seems appropriate and nice.

Additional comments

I ask the author to consider the following comments.

1. The introduction is weak and literature review should be improved. There are some papers which are related to this study and have not been cited. For instance,

a) Hadi-Vencheh, A., & Mohamadghasemi, A. (2013). An integrated AHP–NLP methodology for facility layout design. Journal of Manufacturing Systems, 32(1), 40-45.
b) Niroomand, S., Hadi-Vencheh, A., Şahin, R., & Vizvári, B. (2015). Modified migrating birds optimization algorithm for closed loop layout with exact distances in flexible manufacturing systems. Expert Systems with Applications, 42(19), 6586-6597.
c) Mohamadghasemi, A., & Hadi-Vencheh, A. (2012). An integrated synthetic value of fuzzy judgments and nonlinear programming methodology for ranking the facility layout patterns. Computers & Industrial Engineering, 62(1), 342-348.

2. There are some typo-grammatical errors through the text.
3. What is/are the disadvantage(s) of the proposed method?

---

## Round 0.2 · accepted · Accept

Thank you for submitting the revised manuscript, now it is suitable for publication.

Reviewer 2 ·

Basic reporting

see general comments

Experimental design

see general comments

Validity of the findings

see general comments

Additional comments

The authors have addressed my comments for the previous version. One suggestion is that the new references maybe included in the final version as well.

Reviewer 4 ·

Basic reporting

In this version the author addressed all of my comments, hence I recommend acceptance of this study.

Experimental design

It seems correct and adequate.

Validity of the findings

The findings are correct based on my judgement.